# The Prevalence of Previous Coronavirus Disease-19 in Patients with Pulmonary Thromboembolism and Its Effect on Embolism Severity

**DOI:** 10.3390/jcm14061909

**Published:** 2025-03-12

**Authors:** Nagihan Durmuş Koçak, Nuri Tutar, Gizem Çil, Emine Afşin, Ayşegül Şentürk, Derya Aydın, Buket Mermit, Elif Torun Parmaksız, Mustafa Çolak, Elif Yıldırım, Songül Özyurt, Gülru Polat, Elif Tanrıverdi, İlknur Kaya, Nur Aleyna Yetkin, Elif Yılmazel Uçar, Sibel Doğru, Talat Kilic, Hatice Arzu Uçar, Serdar Berk, Tuğba Çiçek, Emine Ozsari, Gamze Kırkıl, Halil İbrahim Yakar, Ersin Alkılınç, Ali Tabaru, Esra Yarar, Emine Aksoy, Burcu Akkök, Şebnem Emine Parspur, Ercan Kurtipek, Fatih Uzer, Utku Tapan, Dildar Duman, Dursun Tatar, Gülistan Karadeniz, İclal Hocanlı, Özge Oral Tapan, Kadir Canoğlu, Fulsen Bozkuş, Nevra Gullu Arslan, Ömer Tamer Doğan, Mahşuk Taylan, Ayşe Pala

**Affiliations:** 1Department of Pulmonary Medicine, Sancaktepe Şehit Prof. Dr. Ilhan Varank Training and Research Hospital, Istanbul 34785, Türkiye; dreliftorun@yahoo.com; 2Department of Pulmonary Medicine, School of Medicine, Erciyes University, Kayseri 38280, Türkiye; drnuritutar@gmail.com (N.T.); alleynakemik@gmail.com (N.A.Y.); 3Department of Pulmonary Medicine, Faculty of Medicine, Ataturk University, Erzurum 25240, Türkiye; g_i_z_e_m_639@hotmail.com (G.Ç.); eucar1979@yahoo.com (E.Y.U.); 4Department of Pulmonary Medicine, Faculty of Medicine, Bolu Abant İzzet Baysal University, Bolu 14030, Türkiye; emineafsin@yahoo.com (E.A.); dreminedemirok@hotmail.com (E.O.); 5Department of Pulmonary Medicine, Ankara Atatürk Training and Research Hospital, Ankara 06100, Türkiye; asenturk1967@yahoo.com; 6Department of Pulmonary Medicine, Balikesir Atatürk City Hospital, Balikesir 10100, Türkiye; drderyacelebi@gmail.com; 7Department of Pulmonary Medicine, Faculty of Medicine, Van Yuzuncu Yil University, Van 65080, Türkiye; buketmermitcilingir@gmail.com; 8Department of Chest Diseases, Faculty of Medicine, Balikesir University, Balikesir 10145, Türkiye; drmclk@yahoo.com; 9Department of Pulmonary Medicine, Istanbul Sureyyapasa Chest Diseases and Chest Surgery Training and Research Hospital, Istanbul 34844, Türkiye; eky.yil@gmail.com (E.Y.); dremineaksoy95@gmail.com (E.A.); dildaryetis@yahoo.com (D.D.); 10Department of Pulmonary Medicine, Faculty of Medicine, Recep Tayyip Erdoğan University, Rize 53200, Türkiye; drsongul@gmail.com; 11Department of Chest Diseases, Dr Suat Seren Chest Diseases and Surgery Training and Research Hospital, Izmir 35170, Türkiye; gulruerbay@yahoo.com (G.P.); tatar.dursun@gmail.com (D.T.); drglstn35@gmail.com (G.K.); 12Department of Pulmonary Medicine, Yedikule Chest Diseases and Thoracic Surgery Training and Research Hospital, Istanbul 34020, Türkiye; dr.elif06@gmail.com; 13Department of Pulmonary Medicine, Faculty of Medicine, Kütahya Health Sciences University, Kutahya 43030, Türkiye; dr.ilknurcan.lc@gmail.com (İ.K.); adefnenaz@hotmail.com (Ş.E.P.); 14Department of Pulmonary Medicine, Faculty of Medicine, Gaziantep University, Gaziantep 27310, Türkiye; drsibelahmet@hotmail.com (S.D.); mahsuktaylan@gmail.com (M.T.); 15Department of Pulmonary Medicine, School of Medicine, Inonu University, Malatya 44280, Türkiye; talatkilic2013@gmail.com; 16Department of Pulmonary Medicine, Ministry of Health Tokat State Hospital, Tokat 60100, Türkiye; arzuucar45@hotmail.com; 17Department of Pulmonary Medicine, School of Medicine, Cumhuriyet University, Sivas 58140, Türkiye; serdar.berk@yahoo.com (S.B.); tdogangs@gmail.com (Ö.T.D.); 18Department of Pulmonary Medicine, Ministry of Health Konya Numune Hospital, Konya 42060, Türkiye; dr.tugbacicek@hotmail.com; 19Department of Pulmonary Medicine, School of Medicine, Fırat University, Elazığ 23119, Türkiye; gamkirkil@yahoo.com; 20Department of Chest Diseases, Faculty of Medicine, Tokat Gaziosmanpaşa University, Tokat 60250, Türkiye; halil_yakar@hotmail.com; 21Department of Pulmonary Medicine, Ministry of Health Sinop Ataturk State Hospital, Sinop 57000, Türkiye; ersinalkilincmd@gmail.com; 22Department of Pulmonary Medicine, Ministry of Health Söke Fehime Faik Kocagöz State Hospital, Aydin 09200, Türkiye; tabaruali@yahoo.com; 23Department of Pulmonary Medicine, Ministry of Health Kahramanmaras Necip Fazil City Hospital, Kahramanmaraş 46050, Türkiye; yararesra81@gmail.com; 24Department of Pulmonary Medicine, Faculty of Medicine, Kahramanmaras Sutcu Imam University, Kahramanmaraş 46040, Türkiye; bkaraokur@hotmail.com (B.A.); fulsenbatmaz@gmail.com (F.B.); 25Department of Pulmonary Medicine, Konya City Hospital, Konya 42020, Türkiye; kurtipek14@hotmail.com; 26Department of Pulmonary Medicine, Faculty of Medicine, Akdeniz University, Antalya 07070, Türkiye; uzerfatih@gmail.com; 27Department of Pulmonary Medicine, Faculty of Medicine, Mugla Sitki Kocman University, Mugla 48000, Türkiye; drutkutapan@yahoo.com (U.T.); ozgeoral@hotmail.com (Ö.O.T.); 28Department of Pulmonary Medicine, Sanliurfa Mehmet Akif Inan Training and Research Hospital, Sanliurfa 63040, Türkiye; iclalhocanli2163@gmail.com; 29Department of Pulmonary Medicine, Istanbul Sultan 2. Abdulhamid Han Training and Research Hospital, Istanbul 34668, Türkiye; kadircano@gmail.com; 30Department of Pulmonary Medicine, Ministry of Health Samsun Education and Research Hospital, Samsun 55090, Türkiye; nevragullu@hotmail.com; 31Department of Pulmonary Medicine, Kocaeli Derince Training and Research Hospital, Kocaeli 41900, Türkiye; draysepala@gmail.com

**Keywords:** COVID-19, hospital mortality, patient acuity, pulmonary embolism, risk factors

## Abstract

**Background/Objectives**: The association between past coronavirus disease-19 (COVID-19) infection and pulmonary thromboembolism (PTE) is a potential research topic. We aimed to research the prevalence of previous COVID-19 infection in patients with PTE and to determine whether there is a difference in embolism severity in these cases. **Methods:** Study design: Multicenter, observational, cross-sectional. Patients diagnosed with PTE between 11 March 2022 and 11 March 2023 were prospectively included in this study, excluding cases with PTE along with active COVID-19, patients under the age of 18, and pregnant patients. Group 1 consisted of PTE cases with previous COVID-19, and Group 2 consisted of PTE cases without previous COVID-19. Key variables are D-Dimer level, right ventricle/left ventricle (RV/LV) ratio, simplified pulmonary embolism severity score, and treatment type. **Results:** A total of 1185 patients (Group 1; *n* = 360, Group 2; *n* = 825) were included in this study. The proportion of patients with RV/LV ratio > 1 on computed tomography pulmonary angiography (CTPA) was significantly high in Group 2 compared to Group 1 (27.9% vs. 19.7%, *p* = 0.003). In multivariate logistic regression analysis, the absence of any identifiable risk factor for PTE was found to be a 0.46-fold protective factor in the presence of previous COVID-19 (OR: 0.456 95% CI: 0.274–0.760, Wald = 9.070, df = 1, *p* = 0.003) and an RV/LV ratio > 1 on CTPA was found to be a 0.60-fold protective factor (OR: 0.603, 95% CI: 0.365–0.998, Wald = 3.874, df = 1, *p* = 0.049). **Conclusions:** The prevalence of previous COVID-19 infection in PTE cases was 30.4%, and 26.3% of idiopathic cases had previous COVID-19 infection. Although the parameters related to embolism severity were higher in the non-COVID-19 group, multivariate analyses revealed a 2.2-fold increased risk for idiopathic PTE and a 1.7-fold increased risk for RV/LV ratio > 1 on CTPA in patients without COVID-19 compared to those with prior COVID-19.

## 1. Introduction

Venous thromboembolism (VTE), which is clinically recognized as pulmonary thromboembolism (PTE) or deep vein thrombosis (DVT), is the third most common acute cardiovascular syndrome in the world after myocardial infarction and stroke, and studies have reported an increasing trend in its annual incidence [1,2,3]. There are well-defined environmental and genetic risk factors for the development of VTE. Currently, it is thought to occur due to the interaction of patient-related risk factors, which are usually permanent, and predisposing factors, which are usually transient and related to the newly developing condition. Major trauma, previous VTE, hip-knee replacement, and lower extremity fracture are among the strong risk factors. Cancer, chemotherapy, oral contraceptive (OCS) use, or post-partum period are examples of moderate predisposing factors, while pregnancy, obesity, and advanced age are examples of weak risk factors [4].

The disease caused by the Severe Acute Respiratory syndrome coronavirus-2 (SARS-CoV-2) from the coronavirus family has been named Coronavirus disease-19 (COVID-19) [5]. It has been reported that different mechanisms may be effective in the development of coagulopathy susceptibility associated with COVID-19. A cytokine storm (Interleukin-6, Interleukin-1, Tumor Necrosis Factor-Alfa), characterized by an excessive secretion of proinflammatory cytokines as a result of lung damage that may develop during infection, occurs as a result of the binding of SARS-CoV-2 to the alveolar epithelium with the angiotensin-converting enzyme-2 (ACE-2) receptor. These cytokines can promote procoagulation by stimulating numerous processes related to the activation of immune cells in response to changes in the vascular environment. In addition, the activation of the coagulation cascade due to infection and dysfunction of endothelial cells contributes to the development of the thrombotic process. Pulmonary intravascular coagulopathy and thrombin formation are considered to be the main mechanisms in pathogenesis. On the other hand, it has also been reported that compared to pathologies other than COVID-19, endothelial inflammation, platelets, increased procoagulant factors, decreased endogenous anticoagulant factors, and the presence of antiphospholipid antibodies may be more pronounced. In conclusion, immobilization, disseminated intravascular coagulation, and excessive inflammation are held responsible for the development of macrovascular (VTE and arterial thrombi) and microvascular thrombi in patients [6,7,8,9].

Thrombotic complications such as DVT, PTE, myocardial infarction, stroke, and disseminated intravascular coagulation have been reported since the early stages of the pandemic [6,7]. Pronounced prothrombotic changes in severe acute COVID-19 are known to contribute to morbidity and mortality [10]. Thromboembolic cardiovascular risk reportedly remains elevated for at least one year in individuals recovering from severe acute COVID-19 [11]. It has been reported that acute cardiovascular complications after COVID-19 increased significantly in the first 6 months after infection, and arrhythmia and heart failure are also among the complications [12]. Approximately 4% of patients hospitalized with a diagnosis of COVID-19 were found to develop cardiovascular events (stroke, myocardial infarction, and peripheral artery disease) within 3 months after discharge [13]. In a retrospective study, thrombotic events in patients discharged without thromboprophylaxis were ischemic stroke, intracardiac thrombus, segmental PTE, and thrombosed arteriovenous fistula; the incidence rate was reported as 2.5% [14].

Many studies evaluate the association between COVID-19 and PTE in the literature. Most of these studies investigating the frequency of PTE in active COVID-19 disease are retrospective and include heterogeneous patient groups. In different centers, varying degrees of PTE incidence and prevalence have been reported in the presence of COVID-19 infection of different severity [6,7,15,16,17,18]. On the other hand, in our clinical practice, we observe that some patients who present with PTE have had a previous COVID-19 infection despite the absence of other risk factors. Although there are data in the literature that the risk of thromboembolism remains high even months after COVID-19 infection (110 days) [19], further studies are needed on long-term thromboembolic risks in patients who survived COVID-19. The presence of previous COVID-19 may be a predisposing factor for the development of PTE, and determining the underlying risk factors for PTE is important in deciding on the duration of treatment and patient management.

In patients with previous COVID-19, factors such as the severity of the infection, the extent of lung involvement, the need for treatment in hospital or intensive care, and the presence of prophylactic anticoagulant therapy may affect the severity of PTE. Determining whether the severity of PTE varies compared to non-COVID cases may contribute to the determination of the treatment method, the duration of treatment, and patient monitoring.

PTE is a major thromboembolic event with known risk factors, mainly surgery, cancer, immobilization, or genetic predisposition. In this study, we aimed to determine the prevalence of previous COVID-19 among all risk factors by conducting a detailed risk factor questionnaire in PTE patients. Whether there was a difference in terms of embolism severity in PTE patients with previous COVID-19 was determined as a secondary endpoint.

## 2. Materials and Methods

### 2.1. Study Design

Multicenter, observational, cross-sectional.

### 2.2. Patient Selection

For one year, from 11 March 2022, newly diagnosed PTE patients and, in order to rule out active infection, cases with at least 30 days of COVID-19 infection were included in our study. Since there are data from previous studies indicating that the first 30 days are important for the period from COVID-19 diagnosis to the development of thromboembolic events, we included cases in our study that were at least 30 days old after receiving a COVID-19 diagnosis to distinguish it from active infection [20,21,22]. However, patients with active COVID-19 and PTE, under 18, and pregnant cases were excluded. Clinical probability scoring, D-Dimer measurement, Computed Tomography Pulmonary Angiography (CTPA) or lung scintigraphy as imaging methods, and lower extremity compression venous doppler ultrasonography (CUS) were used for the diagnosis of PTE and DVT, respectively. PTE severity was classified as low, intermediate-low, intermediate-high, and high according to the early mortality risk (30 days). Accordingly, cases with a simplified Pulmonary Embolism Severity Index (sPESI) score of 0, no hemodynamic instability, no right ventricular (RV) dysfunction on CTPA or transthoracic echocardiography (TTE), and no increase in troponin levels were considered low risk. Cases with sPESI ≥ 1 and no hemodynamic instability, both negative or one positive of the parameters of RV dysfunction and increased troponin levels, were grouped as intermediate-low risk. sPESI ≥ 1, no hemodynamic instability but high cardiac troponin levels and signs of RV dysfunction were classified as intermediate-high risk. Patients with hemodynamic instability constituted the high-risk group [4]. Group 1 consisted of patients with COVID-19 and newly diagnosed PTE; Group 2 consisted of patients without COVID-19 and newly diagnosed PTE.

### 2.3. Independent Variables

Age, gender, predisposing factors for VTE (defined as strong, moderate, and weak risk factors according to the European Society of Cardiology-ESC 2019 Guidelines for the Diagnosis and Management of Acute Pulmonary Embolism [4]). The time from COVID-19 to the diagnosis of PTE (days), symptoms (hemoptysis, dyspnea, chest pain, syncope, cough, DVT symptoms), laboratory findings (hemoglobin, hematocrit, platelet count, white blood cell count-WBC, D-dimer, troponin (higher/lower than laboratory threshold value), urea, creatinine, proB-type natriuretic peptide (proBNP, higher/lower than laboratory threshold value), diagnostic imaging method (lung scintigraphy, CTPA), thrombus burden on CTPA (unilateral/bilateral), anatomical localization (main pulmonary artery/lobar/segmental/subsegmental), main pulmonary artery/aorta diameter > 1 (present/absent), right ventricle/left ventricle (RV/LV) > 1 (present/absent), TTE findings, if performed bilateral lower extremity CUS findings (no DVT/unilateral thrombus/bilateral thrombus), sPESI score, PTE risk assessment (high/intermediate-high/intermediate-low/low risk), type of treatment (low-molecular weight heparin-LMWH, unfractionated heparin-UFH, vitamin K antagonists-VKAs, non-vitamin K antagonist oral anticoagulants-NOACs, systemic thrombolytics), site of treatment (initial site of treatment when PTE was diagnosed: Intensive care/service/outpatient), length of hospital stay (days), in-hospital mortality (present/absent).

### 2.4. Endpoints

The primary endpoint is the prevalence of previous COVID-19 in patients with PTE. The secondary endpoint is determining whether embolism severity differs in those with a previous COVID-19 diagnosis compared to those without.

### 2.5. Statistics

Descriptive statistics were expressed as frequency, percentage, mean ± standard deviation (SD), and median (minimum-maximum) value. The chi-square test was used to compare the groups according to categorical variables. In the comparison of continuous variables, if the distribution of the variable conforms to the normal distribution, a parametric test (T-Test) was used; if not, a nonparametric test (Mann–Whitney U) was used. Following univariate analyses, multivariate binary logistic regression analysis adjusted for age and sex was performed to identify independent variables associated with PTE severity affecting the dependent variable (presence/absence of prior COVID-19). Odds ratios (ORs), 95% Confidence intervals (CIs), and *p*-values were evaluated with the regression model. For statistical significance level, *p*-value ≤ 0.05 was considered significant. Analyses were performed using the Statistical Package of Social Sciences (SPSS, version 25) software program (IBM® SPSS®, Chicago, IL, USA).

### 2.6. Ethics

The study protocol was prepared in accordance with the Declaration of Helsinki and Good Clinical Practices, and written informed consents were obtained. The trial was approved by the local ethics committee (Approval Date/No: 09.03.2022/E-46059653-020).

## 3. Results

### 3.1. Patient Characteristics and the Prevalence of Previous Coronavirus Disease-19

Forty-four investigators from 31 centers participated in our study. Between 11 March 2022 and 11 March 2023, 1185 patients with a mean age of 64.1 ± 16.3 years, 607 (51.2%) females and 578 (48.2%) males, were prospectively included. At least one predisposing factor for VTE was detected in 1006 (84.9%) patients, while no risk factor was detected in 179 (15.1%) patients (idiopathic). Accordingly, strong risk factor: 341 (28.8%), moderate risk factor: 372 (31.4%), and weak risk factor was present in 293 (24.7%) patients. In addition to all known risk factors, we also questioned the presence of past COVID-19. Patients with previous COVID-19 infection constituted Group 1 (*n* = 360, 30.4%). Those who had not had COVID-19 constituted Group 2 (*n* = 825). The median time from COVID-19 to the diagnosis of PTE was 120.0 (min-max: 30–980) days. Groups 1 and 2 did not differ in mean age and gender (Figure 1).

### 3.2. Risk Factors for Pulmonary Thromboembolism

Among the strong risk factors, hospitalization for heartfailure (HF) or atrial fibrillation (AF)/flatter in the last three months was significantly higher in Group 1 (5.3% vs. 2.8%, *p* = 0.033). Intermediate risk factors included blood transfusion (3.9% vs. 1.9%, *p* = 0.049), congestive HF or respiratory failure (21.9% vs. 17.1%, *p* = 0.048), OCS therapy (5.0% vs. 2.2%, *p* = 0. 009) and infection (21.1% vs. 14.0%, *p* = 0.002) were also significantly higher in Group 1, while the frequency of cancer (11.9% vs. 19.5%, *p* = 0.001) was higher in Group 2. There was no difference between the groups in terms of weak risk factors. A total of 47 (26.3%) of the 179 patients in whom no risk factor could be identified (idiopathic) were in Group 1. However, no significant difference was observed in the distribution of these cases according to groups (13.1% vs. 16.0%, *p* = 0.19) (Table 1).

### 3.3. The Severity of Pulmonary Thromboembolism in Groups

Among the symptoms, cough frequency was higher in Group 1 patients (29.4% vs. 22.5%, *p* = 0.012), and the other symptoms did not differ significantly between the groups. Age-adjusted D-Dimer was elevated in 648 of 686 Group 2 patients and 289 of 323 Group 1 patients (94.5% vs. 89.5%, *p* = 0.004). Serum troponin (*n* = 996) and proBNP (*n* = 507) were not significantly different between the groups. WBC was lower in Group 1 patients (8810 vs. 9420, *p* = 0.043). The most common diagnostic method was CTPA in both groups, and there was no difference between the two groups according to unilateral or bilateral thrombus formation and anatomical localization on CTPA. However, the proportion of cases with RV/LV ratio > 1 on CTPA was significantly higher in Group 2 than in Group 1 (27.9% vs. 19.7%, *p* = 0.003). TTE was performed in 1069 patients, and no difference was found between the groups according to TTE parameters. In a total of 873 patients who underwent CUS, the rate of DVT-free patients was higher in Group 1 (*n* = 285) than in Group 2 (*n* = 588) (73.0% vs. 65.6%, *p* = 0.004) (Table 2).

When we evaluated according to PTE severity, the sPESI score was median in Group 1: 1.0 (min-max: 0–4) and median in Group 2: 1.0 (min-max: 0–5), and the difference was statistically significant (*p* = 0.025). However, no difference was found between the groups according to the PTE risk level, indicating early mortality. In terms of treatment, the proportion of patients receiving systemic thrombolytic drugs was significantly higher in Group 2 (11.3% vs. 7.5%, *p* = 0.048), and the proportion of patients receiving NOACs was significantly higher in Group 1 (32.2% vs. 24.2%, *p* = 0.004). There was also a difference between the groups in terms of the place of initiation of treatment; the proportion of patients who started treatment in the intensive care unit was significantly higher in Group 2 (23.4% vs. 14.7%), and the proportion of patients who started treatment as outpatients was significantly higher in Group 1 (14.2% vs. 9.5%) (*p* = 0.001). Median (min-max) duration of hospitalization in Group 1 patients: 8.0 (1–42) days and in Group 2 cases: 8.0 (1–82) days with no significant difference (*p* = 0.21). In a total of 1056 hospitalized patients (Group 1, *n* = 308, Group 2, *n* = 748), the in-hospital mortality rate did not differ between the groups (3.2% vs. 4.1%, *p* = 0.45) (Table 3).

In summary, among the parameters related to PTE severity, sPESI score, the presence of high D-Dimer, RV/LV ratio > 1 on CTPA, presence of concurrent DVT, and need for thrombolytic therapy were significantly lower in the presence of previous COVID-19, while the rate of patients who started outpatient treatment was higher.

Multivariate binary logistic regression analysis was performed to determine independent risk factors associated with PTE in the presence of previous COVID-19. The absence of any identifiable risk factor for PTE (idiopathic) was found to be a 0.46-fold protective factor (OR: 0.456, 95% CI 0.275–0.760, Wald: 9.070, df: 1; *p* = 0.003) and an RV/LV ratio > 1 on CTPA was 0.60-fold protective factor (OR: 0.603, 95% CI 0.365–0.998, Wald: 3.874, df: 1; *p* = 0.049) (Table 4).

## 4. Discussion

In a total of 1185 prospectively recruited patients with newly diagnosed PTE within one year, the prevalence of previous COVID-19 was 30.4%, in addition to all predisposing factors for VTE. When we compared the groups in terms of embolism severity, which was the secondary endpoint of this study, we found that elevated age-adjusted D-Dimer level, RV/LV ratio > 1 on CTPA, and the presence of concomitant DVT were more common in patients who did not have COVID-19. There was also a significant difference between sPESI scores, and the proportion of patients receiving thrombolytic therapy was higher in the group who did not have COVID-19. On the other hand, outpatient treatment and NOAC use were more common in patients with past COVID-19. The groups were similar in terms of the length of hospitalization and in-hospital mortality. In multivariate analyses, the absence of any identifiable risk factor for VTE (idiopathic) was associated with a 2.2-fold increased risk in those who had not had COVID-19 compared to those who had, and an RV/LV ratio > 1 on CTPA was associated with a 1.7-fold increased risk.

A study in the UK found an increase in the incidence and mortality of thromboembolism during the same periods in 2018 and 2019 compared to the pandemic period. Thromboembolic events accounted for 1.4% of all hospital admissions before the pandemic but 2% during the pandemic. The highest increase among all thromboembolic events was found in venous pathologies, especially PTE [23]. In a meta-analysis, the prevalence of VTE was found to be 23.9% (PE: 11.6%, DVT: 11.9%) in patients hospitalized due to COVID-19 infection and receiving prophylactic anticoagulant treatment and 30.4% in those followed up in intensive care [6]. Increased risk for VTE has also been reported in mild infections that do not require hospitalization. This association is weaker than those with hospital and intensive care admission [24]. As can be seen, there are many studies on the association between concurrent COVID-19 infection and VTE, and the findings differ according to the patient groups included in this study, research centers, and study designs.

On the other hand, it has been shown in various studies that the risk persists after discharge following hospitalization due to COVID-19. In a retrospective observational study, the cumulative incidence of VTE at 30 days following discharge was 0.6% [14], and in another study, the VTE episode rate within 42 days was 4.8 per 1000 discharges [25]. In 2832 adult patients hospitalized with a diagnosis of COVID-19, the incidence of VTE events within 90 days after discharge was 1.3% [26]. Our study found that previous COVID-19 infection was present in approximately one-third of the patients, even at least 30 days after infection. We found the median time from COVID-19 infection to the diagnosis of PTE was 120.0 days. Studies have emphasized especially the first 30 days. For instance, in a study in which 52 cases followed up in different centers were evaluated, the time from diagnosis to the time of PTE after COVID-19 infection was found to be in a wide range (7–180 days) with a mean of 35.1 days. At the same time, most patients did not receive anticoagulant treatment because of mild disease at the time of infection [20]. In the study by Katsoularis et al. using Swedish records, there was an approximately 5-fold increase in the risk ratio for DVT and a 33-fold increase in the risk ratio for PTE within 30 days of infection [21]. In a registry study by Giannis et al. involving 4906 patients, the incidence of VTE within 90 days after discharge was found to be 1.55% [27]. Thromboembolic events developed within 30 days of discharge in 2.0% of four hundred and forty-seven patients [22]; in another cohort study, the incidence of symptomatic VTE was 2.6% within 42 days of discharge in 303 patients [28]. The frequency of post-discharge thrombotic events and follow-up periods vary in studies. In our study, we questioned the presence of previous COVID-19 without differentiating between mild or severe infection and included cases with at least 30 days of history to differentiate from active infection. Thus, we did not find a similarly designed study in the literature.

In a meta-analysis including 16 studies and 5826 patients, including PTE patients with COVID-19, comorbidities associated with PTE in non-COVID patients (pulmonary diseases, obesity, HF, and cancer) did not constitute a significant risk factor for PTE in the presence of COVID-19 [29]. The presence of chronic HF was inversely associated with PTE in COVID-19 patients [17]. We found no significant difference between the groups in terms of other comorbid conditions, although hospitalization due to HF/respiratory failure or HF/AF was found at a higher rate in the group with COVID-19, except for cancer. In our study, the frequency of some moderate risk factors for the development of PTE was higher in the group with COVID-19. However, we think that our sample size may pose a limitation in interpreting the results due to the low number of patients in the groups.

The immune system and the coagulation pathway are closely interrelated. It is thought that viral infections may trigger the development of VTE by activating the systemic inflammatory response and causing an imbalance between the procoagulant/anticoagulant effect [30,31,32]. Since DVT originating from the lower extremities is not frequently associated with COVID-19 patients with PTE, the development of pulmonary microthrombosis originating from local hypercoagulopathy and possibly associated with endothelial cell dysfunction is blamed for the development of PTE in these patients [30]. In one study, the prevalence of DVT was 11% in patients with COVID-19 and PTE [29]. Again, in a series of 5 patients with mild COVID-19, none had symptoms and signs of DVT [33]. In a multicenter, retrospective, and case-control study conducted in Spain, PTE risk factors and leg symptoms were less frequent, and D-Dimer increase was lower in the COVID-19 population [17]. In a retrospective study, age-adjusted D-Dimer levels were not significantly different in COVID-19 and non-COVID patients [18]. Consistent with these results, our study found that the proportion of patients with no DVT was higher in the group with past COVID-19. In addition, the proportion of patients with high age-adjusted D-Dimer was also significantly lower in this group. At this point, we would like to emphasize that in our study, we defined the age-adjusted D-Dimer parameter as a categorical variable above and below the threshold value. Since our study was multicenter and the threshold values and units of D-Dimer, troponin, and proBNP levels differed between centers, the D-Dimer value was entered as above or below the laboratory threshold value. Methodological differences may affect our results. However, considering the studies in the literature that reported lower D-Dimer levels in the coexistence of COVID-19 and PTE, similar to our study, we think that this situation may be related to the prophylactic anticoagulant treatment applied to COVID-19 patients, the severity of COVID-19 infection and the possible less severe VTE that developed in these cases.

Miró et al. reported that smaller pulmonary arteries were affected, and RV dysfunction was similar in patients with PTE associated with COVID-19 [17]. In another study, no significant difference was found between 342 COVID-19 and 147 non-COVID-19 patients on CTPA according to the thrombus location in the pulmonary arteries [18]. We did not find any difference in CTPA according to thrombus localization, but the rate of patients with an RV/LV ratio > 1 was higher in the non-COVID group. Despite the increase in RV/LV ratio on CTPA, there was no significant difference between the groups in TTE. Technical differences in the imaging methods, suboptimal evaluation of TTE, which is a real-time imaging method, or differences in practitioners may have caused this result. On the other hand, the lack of significant difference in cardiac markers between the two groups may have resulted from the decreased sensitivity of the analysis due to the use of these parameters as categorical variables. In addition, among the variables that we thought correlated with embolism severity, the sPESI score and the proportion of patients receiving thrombolytic therapy and hospitalized in the intensive care unit were higher in non-COVID patients. However, in multivariate analyses, only RV/LV ratio > 1 on CTPA was associated with a 1.7-fold increased risk in non-COVID patients. At this point, the severity of infection in patients with COVID-19, the prophylactic anticoagulant treatment given during the disease period, and the duration of this treatment may have affected the severity of PTE. In addition, factors such as changes in the immune response after the infection, microthrombus formation, and involvement of smaller arteries may play a role in the development of less severe PTE in patients with previous COVID-19. Failure to identify any predisposing factor for VTE (idiopathic) was also found to be a risk factor (>2-fold) for those who had not had COVID-19. A previous COVID-19 infection was also observed in approximately one-quarter of idiopathic cases. We think that this is a remarkable finding and further research is needed to determine whether previous COVID-19 infection is a risk factor in cases of idiopathic PTE. Especially since the duration of anticoagulant therapy is one of the difficulties in the management of these idiopathic cases.

There are some limitations to our study. First of all, the small sample size and the lack of a control group that would allow us to determine the severity of previous COVID-19 among all risk factors. In our study, where we included embolism patients for a year and questioned the presence of previous COVID-19, the severity of the infection, the presence of vaccination and thromboprophylaxis, and the lack of consideration of variant characteristics can be considered other limitations. We acknowledge possible selection bias, the retrospective nature of COVID-19 enquiry, and the lack of long-term follow-up after PTE diagnosis. However, in terms of determining the prevalence, which is the primary endpoint, we think that the strengths of our study stand out with its multicenter, inclusion of cases from many different regions, and prospective case recruitment. We believe that future studies are needed regarding the duration of anticoagulant treatment in these cases, as well as the severity of disease and risk factors in PTE developing in the presence of previous COVID-19.

## 5. Conclusions

We concluded that the prevalence of previous COVID-19 infection in patients with PTE was approximately 30%, and interestingly, approximately 25% of idiopathic cases also had a history of COVID-19. There were no differences in the embolism presentation, length of hospitalization, and in-hospital mortality between groups. However, the risk of an increase in RV size on CTPA and being idiopathic was found to be nearly 2-fold increase in PTE cases that had no COVID-19 infection compared to those who had. In addition, the need for systemic thrombolytic therapy was lower, and the rate of initiation of outpatient treatment was higher in PTE patients with previous COVID-19. In light of all our findings, previous COVID-19 may contribute to the development of PTE, and the severity of PTE may be lower in these patients. Therefore, establishing new risk classifications for VTE developing after COVID-19 is important for effective patient management in predicting mortality and determining the treatment approach. However, these findings need to be supported by controlled, large, prospective cohort studies with long-term follow-ups. We think that our study may contribute to the literature as a preliminary study for research that will investigate the cause and effect relationship between previous COVID-19 infection and PTE and its weight among all risk factors.

## Figures and Tables

**Figure 1 jcm-14-01909-f001:**
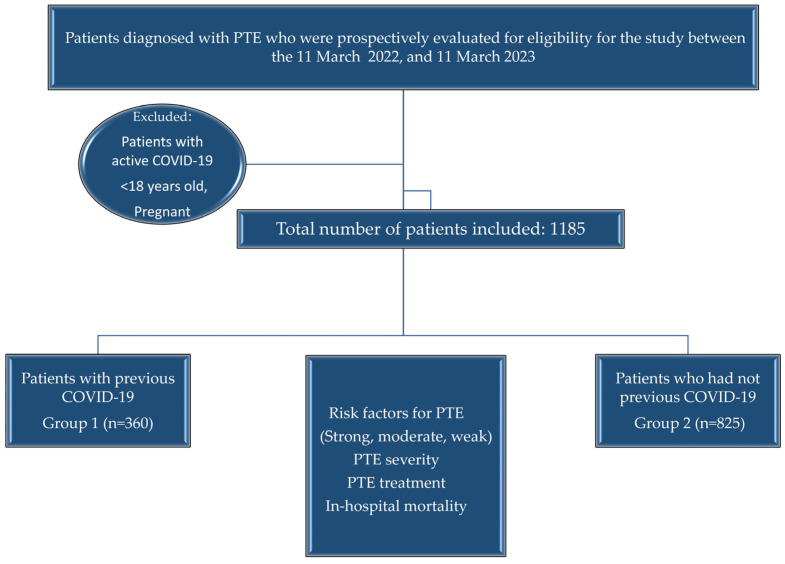
Study Flow Chart.

**Table 1 jcm-14-01909-t001:** Patients’ demographic characteristics and predisposing factors for pulmonary thromboembolism in groups.

Variables	Group 1 (*n* = 360)	Group 2 (*n* = 825)	*p*-Value
Age, mean (SD)	64.1 (16.5)	64.1 (15.7)	0.99 *
Sex, *n* (%)			
Female	190 (52.8)	417 (50.5)	0.48
Male	170 (47.2)	408 (49.5)	
**Strong risk factors (*n* = 341), *n* (%)**			
Fracture of lower limb	11 (3.1)	36 (4.4)	0.29
Hospitalization for HF or AF/flatter (within previous 3 months)	19 (5.3)	23 (2.8)	**0.033**
Hip or knee replacement	14 (3.9)		
Major trauma	7 (1.9)	41 (5.0)	0.42
MI (within previous 3 months)	5 (1.4)	26 (3.2)	0.25
Previous VTE	32 (8.9)	9 (1.1)	0.66
Spinal cord injury	4 (1.1)	68 (8.2)	0.71
**Moderate risk factors (*n* = 372), *n* (%)**			
Arthroscopic knee surgery	7 (1.9)	8 (1.0)	0.17
Autoimmune diseases	10 (2.8)	20 (2.4)	0.72
Blood transfusion	14 (3.9)	16 (1.9)	**0.049**
Central venous lines	7 (1.9)	8 (1.0)	0.17
Intravenous catheters and leads	20 (5.6)	31 (3.8)	0.16
Chemotherapy	22 (6.1)	67 (8.1)	0.23
Congestive HF or respiratory failure	79 (21.9)	141 (17.1)	**0.048**
Erythropoiesis-stimulating agents	2 (0.6)	2 (0.2)	0.39
Hormone replacement therapy	2 (0.6)	5 (0.6)	0.91
In vitro fertilization	0 (0.0)	1 (0.1)	0.51
Oral contraceptive therapy	18 (5.0)	18 (2.2)	**0.009**
Post-partum period	3 (0.8)	8 (1.0)	0.82
Infection	76 (21.1)	115 (14.0)	**0.002**
Inflammatory bowel disease	1 (0.3)	6 (0.7)	0.35
Cancer	43 (11.9)	161 (19.5)	**0.001**
Paralytic stroke	14 (3.9)	28 (3.4)	0.67
Superficial vein thrombosis	4 (1.1)	10 (1.2)	0.88
Thrombophilia	7 (1.9)	12 (1.5)	0.54
**Weak risk factors (*n* = 293), *n* (%)**			
Bed rest >3 days	140 (39.0)	294 (35.6)	0.27
Diabetes mellitus	80 (22.2)	148 (17.9)	0.09
Arterial hypertension	152 (42.2)	334 (40.5)	0.59
Immobility due to sitting	89 (24.7)	214 (25.9)	0.66
Increasing age	119 (33.1)	289 (35.0)	0.51
Laparoscopic surgery	9 (2.5)	29 (3.5)	0.36
Obesity	48 (13.3)	93 (11.3)	0.31
Varicose veins	26 (7.2)	40 (4.8)	0.10
**No identifiable risk factor (*n* = 47), *n* (%)**	47 (13.1)	132 (16.0)	0.19

* *t*-Test. Abbreviations: AF: Atrial fibrillation, HF: Heart failure, MI: Myocardial infarction, SD: Standard deviation, VTE: Venous thromboembolism. *p* values found to be significant according to univariate analysis results are shown in bold.

**Table 2 jcm-14-01909-t002:** Group comparisons according to symptoms, laboratory, and radiological findings.

Variables	Group 1 (*n* = 360)	Group 2 (*n* = 825)	*p*-Value
**Symptoms, *n* (%)**			
Hemoptysis	30 (8.3)	73 (8.8)	0.76
Dyspnea	314 (87.2)	683 (82.8)	0.07
Chest pain	202 (56.1)	438 (53.1)	0.38
Syncope	23 (6.4)	86 (10.4)	0.06
Cough	106 (29.4)	186 (22.5)	**0.012**
DVT symptoms	55 (15.3)	153 (18.5)	0.16
**Laboratory, *n* (%)**			
Age-adjusted D-Dimer (high)	289 (89.5)	648 (94.5)	**0.004**
Troponin (high)	123 (40.7)	301 (43.4)	0.44
ProBNP (high)	111 (64.9)	208 (61.9)	0.51
Hemoglobin (gr/dL), median (min-max)	12.8 (7.4–17.5)	12.6 (4.8–19.4)	0.36 *
Hematocrit (%), median (min-max)	39.0 (24.0–53.6)	39.0 (17.0–59.0)	0.39 *
WBC (count/µL), median (min-max)	8810(1040–91800)	9420(2000–75,300)	**0.043** *
PLT (count/µLx103), median (min-max)	250 (22–819)	244 (198–751)	0.64 *
Urea (mg/dL), median (min-max)	37.0 (8.0–149.0)	38.0 (10.0–215.0)	0.19 *
Creatinine (mg/dL), median (min-max)	0.86 (0.38–5.50)	0.88 (0.07–8.14)	0.32 *
**Diagnosis, *n* (%)**			
CTPA (*n* = 1097)	335 (93.1)	762 (92.4)	
Lung scintigraphy (*n* = 76)	22 (6.9)	54 (7.6)	0.77
Bilateral thrombus (CTPA), *n* (%)	202 (59.2)	455 (57.9)	**0.67**
CTPA thrombus localization, *n* (%)			
Main pulmonary artery	82 (23.8)	226 (28.9)	
Saddle embolism	20 (5.8)	35 (4.5)	
Lobar	89 (25.8)	214 (27.3)	0.13
Segmental	105 (30.4)	229 (29.2)	
Subsegmental	49 (14.2)	79 (10.1)	
Main pulmonary artery/aorta diameter > 1	67 (19.3)	192 (24.5)	0.06
RV/LV ratio > 1 (CTPA), *n* (%)	68 (19.7)	219 (27.9)	**0.003**
**TTE parameters (*n* = 1069), *n* (%)**			
RV/LV ratio > 1	54 (16.2)	144 (19.6)	0.19
D shape	20 (6.0)	70 (9.5)	0.06
Flattened intraventricular septum	12 (3.6)	48 (6.5)	0.06
Tricuspid insufficiency	158 (47.4)	353 (48.0)	0.88
Mobile right heart thrombus	7 (2.1)	8 (1.1)	0.19
Systolic PAP (mmHg), median (min-max)	35.0 (20.0–83.0)	37.5 (15.0–100.0)	0.79 *
TAPSE (cm), median (min-max)	1.9 (0.0–2.7)	2.2 (0.0–2.8)	0.12 *
LVEF (%), median (min-max)	60.0 (30.0–70.0)	60.0 (30.0–66.0)	0.59 *
**Lower limb CUS (*n* = 873), *n* (%)**			
No DVT	208 (73.0)	386 (65.6)	
Unilateral thrombus	65 (22.6)	166 (28.2)	
Bilateral thrombus	12 (4.2)	36 (6.1)	**0.004**

* Mann–Whitney U Test. Abbreviations: BNP: B-type natriuretic peptide, CTPA: Computed tomography pulmonary angiography, CUS: Compression ultrasonography, DVT: Deep vein thrombosis, LV: Left ventricle, LVEF: Left ventricle ejection fraction, PAP: Pulmonary artery pressure, PLT: Platelet, RV: Right ventricle, TAPSE: Tricuspid annular plane systolic excursion, TTE: Transthoracic echocardiography, WBC: White blood cell. *P* values found to be significant according to univariate analysis results are shown in bold.

**Table 3 jcm-14-01909-t003:** Distribution of patients in groups with pulmonary thromboembolism severity, treatment modalities, and in-hospital mortality.

Variables	Group 1 (*n* = 360)	Group 2 (*n* = 825)	*p*-Value
**sPESI score, median (min-max)**	1.0 (0.0–4.0)	1.0 (0.0–5.0)	**0.025** *
**PTE risk stratification, *n* (%)**			
High	24 (6.7)	65 (7.9)	
Intermediate-high	81 (22.5)	207 (25.1)	0.16
Intermediate-low	106 (29.4)	268 (32.5)	
Low	149 (41.4)	285 (34.5)	
**Treatment, *n* (%)**			
LMWH	316 (87.8)	740 (89.7)	0.33
UFH	11 (3.1)	39 (4.7)	0.19
VKAs	119 (33.1)	277 (33.6)	0.86
NOACs	116 (32.2)	200 (24.2)	**0.004**
Thrombolytic treatment	27 (7.5)	93 (11.3)	**0.048**
**Initial treatment place, *n* (%)**			
Intensive care unit	53 (14.7)	193 (23.4)	
Treatment at hospital	256 (71.1)	554 (67.2)	**0.001**
Treatment at home	51 (14.2)	78 (9.5)	
**In-hospital mortality (*n* = 1056), *n* (%)**	10 (3.2)	31 (4.1)	0.45

* Mann–Whitney U Test. Abbreviations: LMWH: Low-molecular weight heparin, NOACs: Non-vitamin K antagonist oral anticoagulants, PTE: Pulmonary thromboembolism, sPESI: Simplified Pulmonary Embolism Severity Index, UFH: Unfractioned Heparin, VKAs: Vitamin K antagonists. *p* values found to be significant according to univariate analysis results are shown in bold.

**Table 4 jcm-14-01909-t004:** Multivariate binary logistic regression analysis showing variables with an effect on the presence of previous coronavirus disease-19.

Variables	Wald	df	*p*-Value	ORs	95% CI for ORs
Age	0.011	1	0.92	1.001	0.989–1.012
Sex	0.062	1	0.80	1.044	0.746–1.460
Cough	1.970	1	0.16	1.298	0.902–1.867
No identifiable risk factor (idiopathic)	9.070	1	**0.003**	0.456	0.275–0.760
Age-adjusted D-Dimer (high)	2.343	1	0.13	0.620	0.337–1.143
RV/LV ratio > 1 (CTPA)	3.874	1	**0.049**	0.603	0.365–0.998
Main Pulmonary artery/Aorta ratio > 1 (CTPA)	1.317	1	0.25	1.340	0.813–2.210
D-shape (TTE)	0.026	1	0.87	0.943	0.463–1.920
Lower limb CUS	4.104	2	0.13		
Lower limb CUS (1)	1.549	1	0.21	0.776	0.520–1.157
Lower limb CUS (2)	3.140	1	0.08	0.491	0.223–1.078
sPESI score	1.747	1	0.19	0.887	0.743–1.060
NOACs	0.635	1	0.42	1.161	0.804–1.676
Systemic thrombolytic treatment	0.487	1	0.49	1.314	0.610–2.829
Initial treatment place	2.025	2	0.36		
Initial treatment place (1)	0.079	1	0.78	0.919	0.509–1.659
Initial treatment place (2)	1.391	1	0.24	0.621	0.282–1.370

Lower limb CUS (1): Unilateral thrombus, CUS (2): Bilateral thrombus. Initial treatment place (1): Hospital, Initial treatment place (2): Intensive care unit. Abbreviations: CI: Confidence Interval, CTPA: Computed tomography pulmonary angiography, CUS: Compression ultrasonography, LV: Left ventricle, NOACs: Non-vitamin K oral anticoagulants, OR: Odds ratio, RV: Right ventricle, sPESI: simplified Pulmonary Embolism Severity Index, TTE: Transthoracic echocardiography. *p* values found to be significant according to multivariate analysis results are shown in bold.

## Data Availability

The raw data presented in this study are available from the corresponding author upon reasonable request. The data are not publicly available due to privacy concerns.

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
