# Peer review of "The Prevalence of Previous Coronavirus Disease-19 in Patients with Pulmonary Thromboembolism and Its Effect on Embolism Severity"

_jcm, 2025, doi:10.3390/jcm14061909_

Round 1

Reviewer 1 Report

Comments and Suggestions for Authors

The study is predominantly descriptive, presenting results and providing a comparative analysis with the existing literature. However, there are several noteworthy aspects that warrant further discussion. For example: 
1)    The D-dimer values in Group 2 were significantly higher compared to Group 1 with COVID-19. Due to COVID-19, there was a considerable increase in the cutoff levels, far exceeding the age-adjusted thresholds. Given that this is a multicenter study, how were the D-dimer cutoff levels determined? Were the laboratory methods for D-dimer detection standardized across all centers, or could methodological differences have influenced the analysis? Additionally, why were the D-dimer levels in Group 2 significantly higher? Shouldn't this be the case for Group 1 instead? Please provide a comment on this.
2)    It is interesting to note that although the RV/LV ratio >1 on CTPA was higher in Group 2, the RV/LV ratio >1 assessed by echocardiography, as well as biomarker levels such as NT-proBNP and troponin, showed no significant differences between the two groups. Please comment on this.
3)    Based on the description of the results, the multivariate binary logistic regression analysis was conducted to identify independent risk factors associated with PTE in patients with a history of COVID-19. Interestingly, the absence of any identifiable risk factor for PTE (idiopathic) was found to be a 0.46-fold protective factor (95% CI 0.275-0.760; p = 0.003), while an RV/LV ratio >1 on CTPA was identified as a 0.60-fold protective factor.Therefore the results indicate that the absence of identifiable risk factors for PTE (i.e., idiopathic cases) was a protective factor, with an odds ratio (OR) of 0.46. This means that individuals without traditional risk factors had approximately a 54% reduction in the likelihood of developing PTE compared to those with known risk factors. Similarly, an RV/LV ratio >1 on CTPA was associated with an OR of 0.60, suggesting that patients with this characteristic experienced a 40% reduction in the risk of PTE. This finding may seem counterintuitive, as an elevated RV/LV ratio is typically indicative of right ventricular dysfunction. Therefore, it is essential to explore and discuss the possible mechanisms underlying this result. Perhaps this discussion could be revised in the conclusion.

Reviewer 2 Report

Comments and Suggestions for Authors

Abstract

• Whether the study aims to determine the prevalence of embolism in patients infected with COVID-19, if there are differences in severity, or if the effects of previous COVID-19 infection share a role in the progression of embolism.

discuss how healtcare workers were the most exposed to the infection and comorbidities. cite doi:10.7416/ai.2021.2439

• Define the study design (multicenter, cross-sectional, observational), inclusion/exclusion criteria, and key variables (RV/LV ratio, D-dimer levels, anticoagulation use).

• Report important p-values and effect sizes for statistically significant differences (eg, increased odds of idiopathic PTE, RV/LV ratio differences).

• Were COVID-19 infection and severity of PTE independent risks salient in multivariate analysis?

• Measures the clinical equalization impact of ‘previous COVID-19 was common in patients with PTE’ abstraction statements™ by emphasizing how its presence changes clinical presentation and risk stratification and treatment response.

Introduction

Finish with introduction of PTE as a major thromboembolic event with known risk factors (cancer, immobilization, surgery, genetic predisposition)

• Explain potential hypercoagulability related to COVID-19 and prior research on the risk of thrombosis after COVID-19

• Introduce (most likely the missing) evidence gap: Data on long-term thromboembolic risks in patients who have survived COVID is scarce (particularly with ill-defined idiopathic PTE cases).

• Explain why this study is warranted in light of existing data that describes acute COVID-associated VTE. cite doi:10.3390/hematolrep15020024. 

• Explain the clinical relevance of comparing PTE severity in COVID and non-COVID patients

• Quotes studies from 2022-2024 on occurrence of post COVID thromboembolic events and long-term risk assessment.

Materials and Methods

• Inclusion: PTE patients newly diagnosed.

• Inclusion: Non-pregnant with active COVID-19, age ≥18 years

• Explain why at least 30 days had to elapse after COVID to preclude inclusion (to avoid acute COVIDassociated VTE).

• Describe methods of confirming PTE diagnosis (CT pulmonary angiography, Doppler ultrasound for DVT, clinical risk scoring).

• Specify how embolism severity was evaluated (RV/LV ratio >1, main pulmonary artery/aorta diameter, simplified Pulmonary Embolism Severity Index - sPESI score).

• Choose both parametric (t-tests) and non-parametric (Mann-Whitney U) tests according to data distribution.

• Describe adjustment for confounding variables in the multivariate logistic regression.

• State whether corrections for multiple comparisons (e.g. Bonferroni) were made to prevent false positive findings.

Results

• Clearly summarize how previous COVID-19 influenced on PTE severity (i.e.: lower RV/LV ratio, lower D-dimer lower, increased idiopathic PTE).

• Present adjusted odds ratios (ORs) and confidence intervals (CIs) for multivariate analyses.

• P-values should be standardized in formats (p = 0.003 and not p=0.003)

• Report effect sizes consistently across tables and text.

Differences in RV/LV ratio — does it median RV/LV difference translate into difference in prognostic / mortality risk?

• If idiopathic patte was more prevalent in post-COVID cases, discuss potential underling mechanism.

Discussion

• Describe the key differences in statistics between groups (e.g., prevalence, severity metrics, treatment differences, multiorgan comorbidities ). cite doi:10.1007/s00405-021-06958-4

In comparison with earlier multiple cohort studies, this was one of the first studies which also had a number of advantages: it is one of the largest studies of cohorts free from bias in the controls selected, and the multivariate match allows estimation of the individual impact of different outcome measures.

• Compare data with previous COVID-19 related VTE studies.

• Explain what this study contributes to the post- COVID hypercoagulability literature

• Comment on possible mechanisms for less severe embolism in patients with prior COVID-19:

• Long-lasting anticoagulation effects?

• Changed immune response after infection?

• Endothelial dysfunction from SARS-CoV-2 may cause long-lasting prothrombotic states.

• Describe if COVID-19 history should be treated as an independent risk factor among VTE risk stratification

• Acknowledge selection bias, being retrospective, and not including long-term follow up.

• Propose future studies evaluating duration of anticoagulation in PTE patients post-COVID.

Conclusion

• Rather than “prior COVID-19 was prevalent in PTE cases,” instead describe how this affects presentation and treatment of embolism.

• Main clinical tuvo takehome messages

• Emphasize that previous COVID-19 may contribute to PTE, but the grope of its effect on severity is nuanced

• Stress the importance of new risk stratification models with post-COVID thrombotic risk.

Comments on the Quality of English Language

None

Round 2

Reviewer 1 Report

Comments and Suggestions for Authors

The authors have addressed all the reviewers' questions point by point and have also revised and rewritten selected sections of the manuscript in accordance with the reviewers' recommendations. The manuscript is now more concise and clearer, improving its suitability for publication.